# The protein and volatile components of trail mucus in the Common Garden Snail, *Cornu aspersum*

Kaylene R. Ballard[1,2], Anne H. Klein[1,2], Richard A. Hayes[3], Tianfang Wang[1,2], Scott F. Cummins[1,2]*

1 Genecology Research Centre, University of the Sunshine Coast, Maroochydore DC, Queensland, Australia, 2 School of Science, Technology and Engineering, University of the Sunshine Coast, Maroochydore DC, Queensland, Australia, 3 Forest Industries Research Centre, Forest Research Institute, University of the Sunshine Coast, Maroochydore DC, Queensland, Australia

* scummins@usc.edu.au

**Data Availability Statement:** In order to facilitate the access and use of the C. aspersum transcriptome sequencing data, the raw data in the FASTQ format were deposited in the Sequence

## Abstract

The Common or Brown Garden Snail, *Cornu aspersum*, is an invasive land snail that has successfully colonized a diverse range of global environments. Like other invasive land snails, it is a significant pest of a variety of agricultural crops, including citrus, grapes and canola. *Cornu aspersum* secretes a mucus trail when mobile that facilitates locomotion. The involvement of the trail in conspecific chemical communication has also been postulated. Our study found that anterior tentacle contact with conspecific mucus elicited a significant increase in heart rate from 46.9 to 51 beats per minute. In order to gain a better understanding of the constituents of the trail mucus and the role it may play in snail communication, the protein and volatile components of mucus trails were investigated. Using two different protein extraction methods, mass spectrometry analysis yielded 175 different proteins, 29 of which had no significant similarity to any entries in the non-redundant protein sequence database. Of the mucus proteins, 22 contain features consistent with secreted proteins, including a perlucin-like protein. The eight most abundant volatiles detected using gas chromatography were recorded (including propanoic acid and limonene) and their potential role as putative pheromones are discussed. In summary, this study has provided an avenue for further research pertaining to the role of trail mucus in snail communication and provides a useful repository for land snail trail mucus components. This may be utilized for further research regarding snail attraction and dispersal, which may be applied in the fields of agriculture, ecology and human health.

## Introduction

The Common Garden Snail, *Cornu aspersum* (previously known as *Helix aspersa*), is an invasive land snail and agricultural pest of global significance [1]. A pulmonate gastropod of the Phylum Mollusca, its population has spread from its native Europe to most continents. Its ability to adapt to a wide variety of environments can be attributed to several factors. During hot

Read Archive (SRA-NCBI) database with accession numbers: SRX10567815 (foot), SRX10567816 (anterior tentacle), SRX957716 (mucous gland), SRX2546515 (central nervous system) and SRX1015093 (posterior tentacle).

**Funding:** Grains Research Development Corporation, Contract code is USU1903-001RSX The GRDC has provided funding for this project; however, they had no role in study design, data collection and analysis, decision to publish or preparation of the manuscript.

**Competing interests:** The authors have declared that no competing interests exist. The commercial funding provided by the GRDC does not alter our adherence to PLOS ONE policies on sharing data and materials.

or dry periods, the snail will aestivate in order to prevent desiccation. Similarly, when conditions are cold, the snail will hibernate. The ability to secrete a mucus epiphragm to create a favourable microclimate inside its shell allows *C. aspersum* to survive adverse environmental conditions. In addition, *C. aspersum* is a simultaneous hermaphrodite, which means it exchanges egg and sperm simultaneously when mating [2]. This allows it to mate with any other conspecific, significantly improving the chances of finding a mating partner. A snail of this species can mate up to 6 times in one breeding season and continue to reproduce the following season [1].

As a result of its adaptability, *C. aspersum*, along with several other land snail species, are significant agricultural pests. It is a serious pest of citrus orchards in California [3] and wine growing areas of South Africa and Australia [2, 4]. In addition to the consumption of crops, it has also been implicated as a vector of plant pathogens, including the pathogenic fungus-like organism, *Phytophthora citrophthora*, also known as branch canker, on citrus trees [5]. Current control strategies for land snails include baiting with toxic molluscicides, commonly methiocarb and metaldehyde [6–9], which can have a detrimental effect on native wildlife and crops [10]. Baiting methods are also expensive and thus economically unsustainable [6, 9]. Mechanical methods, such as rolling, cabling and slashing of the crop are commonly employed as an alternative, or an adjunct to molluscicides. These methods effectively displace the snails from the safety of the crop stem, and can lead to snail mortality due to desiccation on the hot ground, particularly when the temperature exceeds 34 ˚C. Another commonly employed technique is burning before sowing. However, this can lead to soil erosion and depletion of organic matter, along with threats to native wildlife, and is unsustainable as a long-term solution [6, 9]. Biocontrol methods for land snails have, to date, been relatively unsuccessful, and the search continues for an effective, sustainable control strategy. Knowledge of land snail communication may contribute to the development of a sustainable and effective control method.

Chemical communication has been the topic of much research in agricultural pest control over the last fifty years, particularly in relation to pheromones. Pheromones are substances that are secreted by an individual, which can be received by another individual of the same species, producing a specific reaction [11]. Globally, over one million hectares of agricultural land are managed using pheromones that lead to mating disruption, and even more hectares using pheromone lures, primarily for insect pests [10]. This is largely driven by the rise in organic farming and pressure to reduce pesticide use for environmental and health reasons. There is also the crucial issue of food insecurity, which hastens the need for effective and safe control of agricultural pest species [10]. Pheromones as pest control agents are attractive for a number of reasons including the small volume required to be effective, species specificity and the indication that they are generally safe for other wildlife [10]. Successful control of agricultural insect pests such as the light brown apple moth, *Epiphyas postvittana*, and the potato moth, *Tecia solanivora* has been achieved through disruption of the chemical communication associated with mating [12, 13]. In other insect orders, there has been similar success, for example in the red palm weevil, *Rhynchophorus ferrugineus*, using pheromone lures to trap males [14].

Like other gastropods, *C. aspersum* secretes a mucus trail when mobile, which has been shown to facilitate several functions, including locomotion. Trail following has been observed for the purpose of reducing the amount of mucus required by the following snail, and possibly as a source of nutrition [15, 16] and may also function in chemical communication [16, 17]. Trail following as a means of mate finding is well documented in marine [16, 18], and aquatic snails [19]. Trail following behaviour has also been documented in terrestrial gastropods, such as the predatory Rosy Wolfsnail, *Euglandinea rosea* [20], the White-lipped Globe Snail, *Mesodon thyroidus* [21], and the micro snail, *Vallonia excentrica* [22]. While there is still limited research available regarding trail following in land snails, a recent study investigated this

behaviour in five tree snail species endemic to Hawaii. The results found that trails were followed by conspecifics between 66.7% (*Auriculella diaphana*) and 94.1% (*Portulina variablilis*) of the time [23]. These results suggest that trail mucus contains molecules that are detectable by conspecifics.

The identity of the detectable mucus molecules remains unknown; however, they may be proteinaceous and detected via direct contact using paired anterior tentacles. Heart rate variability can reflect responses to internal and external stimuli. For example, predator odours and sex pheromones are associated with an increase in mammalian heart rate [24–26] and this has also been demonstrated in moths [27]. Proteins that function as chemosensory cues have not been described in terrestrial snails, yet have been reported to function as pheromones in several aquatic mollusc species including the gastropods *Aplysia sp.* [28–33], *Biomphalaria glabrata* [34] and the bivalve Silver-Lip Pearl oyster, *Pinctada maxima* [35]. Alternatively, they may be volatile organic compounds (VOCs) that are detected by the larger paired posterior tentacles, which are positioned at the top of the head [36]. A single study has identified a number of different VOCs from the trail mucus of 3 species of land snails [37]. Differences between chemosensory capabilities in *C. aspersum* and the predatory *E. rosea* have previously been described [38], which suggest that *C. aspersum* has a stronger reliance on volatile chemosensory cues than its carnivorous counterpart. The elongated lip extensions of *E. rosea* allow this snail to detect water-soluble components left in the mucus trail of prey snails. However, as *C. aspersum* is primarily herbivorous, it responds to VOCs from plants that represent a food source [39, 40].

Chemical communication research on land snails is scant, and while trail-following studies suggest that land snails communicate via their mucus trail, a pheromone has yet to be identified. The availability of *C. aspersum*, its pest status and relatively large size leads it to be an ideal model organism for research pertaining to olfactory communication in terrestrial snails and slugs. This project investigated the protein and volatile components of *C. aspersum* trail mucus with a view to broadening the knowledge of mucus components, thereby laying the groundwork for identification of one or more putative pheromones that may be utilised in developing a sustainable and effective control method for pest land snails.

## Methods

### Animals and maintenance

All use of animals for this research was carried out in accordance with the recommendations set by the Animal Ethics Committee, University of the Sunshine Coast. Adult garden snails, *C. aspersum*, were obtained from the commercial supplier Glasshouse Gourmet Snails, situated on the Sunshine Coast of Queensland, Australia (-26.90˚S, 152.93˚E) in September 2017 and January 2018. Snails were kept in wooden boxes covered in shade cloth and kept at a controlled temperature of 21–23˚C. Humidity was not controlled and thus varied widely (38–90% relative humidity). Additional snails were sourced in May 2018 from a private garden in Kilcoy QLD (-26.94˚S, 152.56˚E). Snails were fed on a diet of carrot and lettuce, with cuttlefish bone to provide calcium.

### Heart rate assay

Snails were secured and backlit with a Moon Xpower600 light source. Heart rate was counted manually for 60 s only after the snail's anterior tentacles made contact with a clean glass slide. Heart rate was then counted for a further 60 s upon exposure to conspecific trail mucus (n = 18). Care was taken to ensure that no snail was presented with its own trail, by rotating the snails so that the test snail became the next trail layer. As negative control, snails were

presented with Milli-Q water (n = 20), and as a positive control, On-guard Snail Gel (STV International Defenders) was used (n = 22). On-guard Snail Gel is described as a colorless, odour-free slug gel that creates a natural, poison-free barrier for slug and snail. Differences between pre- and post-contact heart rate were compared using a between-measures t-test (IBM SPSS Statistics Version 24).

## Total RNA extraction, sequencing and de novo assembly

Three sections (~5 mm$^3$) were cut with a scalpel from the anterior, mid and posterior sections of the sole of the foot of one adult *C. aspersum*. Tissue was placed in 3 separate 1.5 mL micro-fuge tubes tubes and weighed to ensure a mass between 50 and 100 mg. Tissue was then finely sliced with a clean scalpel blade to facilitate breakdown of muscle tissue, and reweighed. Total RNA was subsequently isolated with the Trizol reagent (Ambion) according to the manufacturer's instructions. The yield and purity of RNA was determined using a Nanodrop spectrophotometer 2000c (Thermo Scientific, Waltham, MA, USA) at 260 and 280 nm. To determine the RNA integrity number (RIN), samples were analysed with an Agilent Bioanalyzer 2100 (Agilent Technologies, USA) to ensure each sample used had a 28S:18S greater than 1.5 and RIN greater than 7. High quality total RNA was pooled from anterior, mid and posterior sections and sent to the Australian Genome Research Facility (Australia), for cDNA synthesis using a cDNA Rapid Library Preparation Kit (Roche, Mannheim, Germany) and subjected to Illumina HiSeq 2500 sequencing (Illumina, San Diego, CA, USA). Additionally, the anterior tentacles were collected from 6 adult snails and subjected to the procedure above.

Clean data (clean reads) were screened from raw sequencing reads based on (1) discard reads with adaptor contamination; (2) discard reads when uncertain nucleotides constituted more than 10% of either read ($N > 10\%$); and (3) discard reads when low quality nucleotides (base quality less than 20) constituted more than 50% of the read. Quality reads were *de novo* assembled using SOAPdevono2 (CLC genomics workbench, version 10.1, Qiagen, Hilden, Germany) with parameters set as follows: seqType, fq; minimum kmer coverage = 4; minimum contig length of 100 bp; group pair distance = 250. Estimation of transcript expression was performed using the *de novo* RNA-Seq analysis tool on the CLC Genomic workbench software with default parameters. Sequence datasets have been deposited in the NCBI Sequence Read Archive (SRA) database (Accession numbers: SRX10567815 and SRX10567816). A transcriptome-derived protein database was prepared by combining the foot and anterior tentacle with tissue transcriptomes of the mucous gland (SRX957716), central nervous system (CNS) (SRX2546515), and posterior tentacle (SRX1015093) using the CLC genomics Workbench and protein sequences predicted using ORF predictor (http://bioinformatics.ysu.edu/tools/OrfPredictor.html).

## Trail mucus collection and protein extraction

Adult *C. aspersum* were washed in water and allowed to crawl in individual glass Petrie dishes for approximately 5 mins. Two methods were chosen to help identify trail mucus proteins. In the first method, eight individual snails produced trail mucus for analysis. Petri dishes were then washed with 10 mL of 0.1% trifluoracetic acid (TFA) and this solution was transferred into 15 mL tubes. Tubes were frozen at -20˚C until required. Snails reproductive systems were examined to ensure maturity. Trail mucus biomolecules were isolated using Sep-Pak plus C18 cartridges (Waters) prepared with acetonitrile, according to the manufacturer's instructions. Biomolecules were eluted with 60% acetonitrile and lyophilised in a Thermofisher Speedvac Concentrator (SC250EXP).

In the second protein extraction method, mucus was collected from 10 snails on 3 separate occasions for a total of 30 snails. Snails were allowed to crawl on a glass sheet measuring 30 × 15 cm for 10 min. Mucus was collected into a 15 mL tube with the aid of a razor blade and a small amount of Milli Q water. Tubes were centrifuged for 5 min at 16,900 xg then for each sample the pellet and supernatant were placed into separate tubes. Proteins were size fractionated by 1D sodium dodecyl sulfate-polyacrylamide gel electrophoresis (SDS-PAGE) using 20 μL of sample in a Mini-Protean TGX (BioRad) gel. All visible bands were excised and prepared for liquid chromatography tandem mass spectrometry (LC-MS/MS) following the protocol of Wang et al [41].

## In-solution digestion of mucus proteins and uHPLC tandem QTOF MS/MS analyses

Lypholised proteins were reconstituted with 50 μL MilliQ water, and concentration was checked using a Nanodrop spectrophotometer 2000c (Thermo Scientific, Waltham, MA, USA) at 260 and 280 nm. Proteins were digested using the protocol described in Hall et al [42] and dividing reagent volumes by 3, to account for the difference in protein concentration. Digested proteins were stored at -20°C to await LC-MS/MS analysis.

Tryptic peptides were resuspended in 100 μL 0.5% formic acid in MilliQ water and analysed by LC-MS/MS on an ExionLC liquid chromatography system (AB SCIEX, Concord, Canada) coupled to a QTOF X500R mass spectrometer (AB SCIEX, Concord, Canada) equipped with an electrospray ion source. Twenty microlitres of each sample was injected onto a 100 mm × 1.7 μm Aeris PEPTIDE XB-C18 100 uHPLC column (Phenomenex, Sydney, Australia) equipped with a SecurityGuard column for mass spectrometry analysis. Solvent A consisted of 0.1% formic acid (aq) and solvent B contained 100% acetonitrile/0.1% formic acid (aq). Linear gradients of 5–35% solvent B over 10 min at 400 μL/min flow rate, followed by a steeper gradient from 35% to 80% solvent B in 2 min and 80% to 95% solvent B in 1 min were used for peptide elution. Solvent B was held at 95% for 1 min for washing the column and returned to 5% solvent B for equilibration prior to the next sample injection. The ionspray voltage was set to 5500 V, declustering potential (DP) 100V, curtain gas flow 30, ion source gas 1 (GS1) 40, ion source gas 2 (GS2) 50 and spray temperature at 450°C. The mass spectrometer acquired mass spectral data in an Information Dependent Acquisition, IDA mode. Full scan TOFMS data was acquired over the mass range 350–1400 amu and for product ion ms/ms 50–1800 amu. Ions observed in the TOF-MS scan exceeding a threshold of 100 cps and a charge state of +2 to +5 were set to trigger the acquisition of product ion. The data was acquired and processed using SCIEX OS software (AB SCIEX, Concord, Canada).

## Protein identification

LC-MS/MS data were imported into the PEAKS studio (Bioinformatics Solutions Inc., Waterloo, ON, Canada, version 7.0) with the assistance of MS Data Converter (Beta 1.3, http://sciex.com/software-downloads-x2110). The database used was a combined transcriptome of the foot and anterior tentacle created in this study and the *C. aspersum* central nervous system, mucous gland and posterior tentacle databases previously created at USC. *De novo* sequencing of peptides, database search and characterising specific post-translational modifications (PTMs) were used to analyse the raw data; false discovery rate (FDR) was set to ≤ 1%, and [-10*log(p)] was calculated accordingly where p is the probability that an observed match is a random event. The PEAKS used the following parameters: (i) precursor ion mass tolerance, 0.1 Da; (ii) fragment ion mass tolerance, 0.1 Da (the error tolerance); (iii) tryptic enzyme specificity with two missed cleavages allowed; (iv) monoisotopic precursor mass and fragment ion

mass; (v) a fixed modification of cysteine carbamidomethylation; and (vi) variable modifications including lysine acetylation, deamidation on asparagine and glutamine, oxidation of methionine and conversion of glutamic acid and glutamine to pyroglutamate.

## Gene protein annotation and expression

To identify the corresponding gene sequences, BLASTp was searched using the precursor protein sequences as a query with an e-value threshold of $1 \times 10^{-3}$. Protein sequences were annotated using SignalP 4.0 (http://www.cbs.dtu.dk/services/SignalP/) to identify protein-coding genes containing a predicted signal peptide sequence (35). NeuroPred (http://neuroproteomics.scs.illinois.edu/cgi-bin/neuropred.py) was employed to predict cleavage sites (>0.5 probability), posttranslational modifications, and bioactive peptide products. Relative gene expression of proteins in the foot, mucous gland, central nervous system and posterior and anterior tentacles of *C. aspersum* was recorded using a reference database comprising the transcriptomes of the aforementioned tissues. This was based on transcripts per kilobase million mapped reads (TPM), utilizing the *de novo* RNA-seq CLC Genomic Workbench 11 software. A heatmap representing z-score relative expression was constructed using an R package (gplots version 3.0.3).

## Sequence alignments and phylogenetic tree construction

The perlucin amino acid sequence identified from the *C. aspersum* trail mucus was aligned against homologs collated from known mollusc perlucin and perlucin-like sequences from the NCBI non-redundant database. An additional *C. aspersum* perlucin-like protein was identified by BLASTp analysis of the transcriptome prepared in this paper (contig 36557). The MEGA-X platform (version 10.1.5) was used for alignments utilising the ClustalW method and a phylogenetic tree constructed using the Neighbour-Joining method [43] A multiple sequence alignment schematic was generated using the LaTeX's TeXShade package [44].

## Mucus collection and VOC extraction using thermal desorption

Six adult *C. aspersum* were allowed to crawl on a glass sheet (30 × 15 cm) cleaned with 80% ethanol, until sufficient mucus was laid down. Approximately 1 mL of mucus was collected into a 1.5 mL microtube with the aid of a razor blade and scalpel, both cleaned with 80% ethanol. The process was repeated 3 times with different snails at different time points. Mucus was transferred into a clean 100 mL Schott bottle with a modified lid, with a 0.45 μm filter attached (Millipore Millex HP). Volatile compounds were trapped onto three thermal desorption tubes (Markes) in series packed with Tenax TA 35/60, Carbograph 1TD 40/60 (344.6 ± 0.748 mg) attached to the outlet, over a period of 6 h, using a vacuum pump (Ilmvac) with a flow rate of approximately 200 ml/min, resulting in a total air sampled of 72L. This process was repeated using an empty container as a negative control, and 'Pheromone' cologne (Pherlure®) as a positive control. After collecting volatiles, tubes were thermally desorbed at 280˚C (Markes, TD-100) and analysed with a gas chromatograph (GC) (Agilent 6890 Series) coupled to a mass spectrometer (MS) (Agilent 5975) and fitted with a silica capillary column (Agilent, model HP5-MS, 30 m × 250 μm ID × 0.25 μm film thickness). GC conditions for acquiring data were–inlet temperature: 250˚C, carrier gas: helium at 51 cm/s, split ratio 13:1, transfer-line temperature: 280˚C, initial temperature: 40˚C, initial time: 2 min, rate: 10˚C/min, final temperature: 260˚C, final time: 6 min. The MS was held at 280˚C in the ion source and the scan rate kept was 4.45 scans/s. Tentative identities were assigned to peaks with respect to the National Institute of Standards and Technology mass spectral library. Mass spectra of peaks from different samples with the same retention time were compared to ensure that the

compounds were indeed the same. Negative control results were compared with mucus results and common substances removed. The eight most abundant volatiles were noted and discussed.

## Results and discussion

### Heart rate in response to conspecific trail mucus

In this study, *C. aspersum* heart rate was investigated to assess whether contact (via anterior tentacle) with conspecific trail mucus could elicit a physiological response (Fig 1, inset). Negative controls (Milli-Q water) resulted in no significant difference in heart rate $t$ (19) = 1.099, $p$ = 0.286 (Base HR—68.7 ± 3.05 (SEM) beats /min, test HR—69.75 ± 2.74 (SEM) beats/min (Fig 1). Positive control (On-guard snail gel) resulted in a significant increase in heart rate, $t$ (21) = 6.513, $p<0.001$ (Base HR– 46.27 ± 1.87 (SEM) beats /min, test HR– 52.59 ± 2.0 (SEM) beats/min). Similarly, upon contact with conspecific trail mucus, there was a significant increase in heart rate $t$ (17) = 4.174, $p$ = 0.001, (Base HR– 46.94 ± 1.31 (SEM) beats /min, test HR– 51.00 ± 1.30 (SEM) beats/min). It should be noted that the negative control baseline was higher than that in the other conditions. This was due to two of the snails in that group having an unusually high heart rate. As comparisons were within groups, it was decided not to eliminate these snails from the calculations.

The observed increase in heart rate is an indicator of autonomic nervous system activation, which supports the idea that snails are responding to detectable elements in, or released, from the mucus. However, it does not demonstrate whether this response was due to excitement resulting from the presumed proximity of another snail, or stress or fear, potentially increasing oxygen and nutrients required for locomotion. The lack of a significant difference in mean heart rate before and after exposure to water suggests that *C. aspersum* is indeed responding to the trail mucus of a conspecific. While these results do not conclusively show that *C. aspersum*

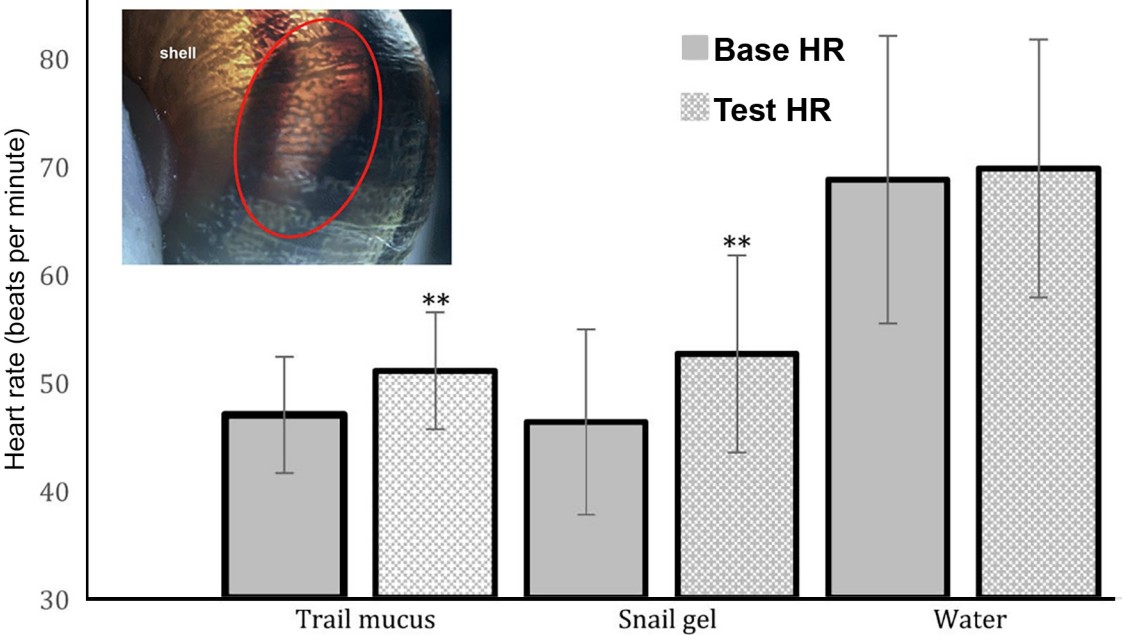

**Fig 1. Mean heart rate before and after exposure to trail mucus, snail gel and water.** ** indicates significant difference ($p<0.05$) using a repeated-measures *t*-test between base and test heart rate.

is attracted to the trail of a conspecific, it does suggest that trail mucus contains components that convey information to the receiving snail. This suggests that the detectable element may be a water-soluble substance, such as a protein and/or VOC. An increase in heart rate has been demonstrated as a response to pheromones or predator odours, in animals as diverse as bulls (26) and moths (27).

### Proteomic analysis of trail mucus

Proteomic analysis was performed on trail mucus obtained from 38 individual *C. aspersum*. A total of 175 proteins were identified (S1 File), of which 29 proteins had no significant similarity to other sequences present in the NCBI non-redundant protein sequence database and 22 proteins were predicted to be secretory (S2 File), with 10 being full-length. Fig 2 provides an annotated summary of the 10 secreted full-length proteins, including the locations of signal peptide, cysteine residues, putative glycosylation sites and cleavage sites. Of these, two have no significant matches, with another found in a related species but with no clear function. RNA-seq quantitative analysis was performed using data available for the snail CNS, mucous gland, foot, anterior and posterior tentacles (Fig 3A). This indicates that the snail's foot and posterior tentacle were relatively abundant in corresponding transcripts, although 3 different genes are more abundant in the CNS and anterior tentacle, and two genes in the mucous gland (Fig 3B). While few proteins demonstrated exclusive tissue expression, several genes were far more abundant in some tissues, such as contigs 104, 129, 936 and 2548 in the foot.

Some proteins we report in this study have previously been reported in *C. aspersum* trail mucus, specifically three proteins reported by Pitt et al [45] that show antimicrobial activity [i.e. two variants of epiphragmin (contigs 129 and 2548) and the uncharacterised contig 3728]. These proteins were all expressed at relatively high levels in the foot (Fig 3C). As the major protein of the epiphragm, a specialised mucus that is required for snail aestivation and hibernation [46], epiphragmin may form an important barrier to help protect the snail from pathogens. One epiphragmin identified (contig 2548) was exclusively expressed in the foot. Its foot-specific expression is consistent with the findings of Campion [47], who has described cells at the ventral area of the foot of *C. aspersum* that contain mucus.

Several trail mucus proteins identified here have been characterized in other species, including the metalloproteinase-like ADAM family protein (contig 2928) and perlucin-like protein (contig 199233). The metalloproteinase-like ADAM family protein has a predicted role in protein degradation, particularly in relation to the extracellular membrane of the cell. However, many functions of this family of proteins remain unclear [48]. Perlucin, although never previously reported in land snail mucus, is well known to be a shell matrix protein of aquatic molluscs [49]. Specifically, it has been shown to enhance the crystallisation of calcium carbonate [50]. However, other studies have suggested that the C-lectin binding domain present in this protein plays a role in linking the living tissue to biomaterial, such as adhesive mucus or byssal thread [51, 52]. It is unclear whether this protein plays a similar role in land snail mucus. Smith et al also identified a perlucin-like protein in the adhesive glue from the dorsal surface of the Dusky Slug *Arion subfuscus*, suggesting that this protein may play a role in defence [53]. Indeed, we found that perlucin-like proteins are widely distributed throughout molluscs, all of which contain six spatially conserved cysteine residues. Our reference *C. aspersum* transcriptome database contained two perlucin-like contigs (including one encoding the trail mucus perlucin-like protein), which is consistent with its apparent diversification within other molluscan species, from gastropods to bivalves. Phylogenetically, *C. aspersum* perlucin-like proteins form a clade with land gastropods (*Achatina fulica* and *Meghimatium fruhstorferi*), but also bivalves (Fig 4A). Based on the number of perlucin-like proteins in *A. fulica*, we

**Contig 1270 - Mucus protein *Helix aspersa* (1e-42)**

MLKKQFILFVAVLLSNAGAVAGNCVNAATSEGSRTLLDGETACAPLDPGETEADKGKRFVTCDNGIIYSQSCGSALRYHAASKICTWPESASCYLAGGDTTDSTTKVPT
PSEPRTTNPPAASLKSTRKPSTQFYTILTTTKPGNQIQPEGPCGPATCKLPDCFCFGARPNLPLVDTPQFIMLTFDDAVTSTVYNGFFKSLLVDNTFKLSNPGGCNIRS
AFYINHDYTDYNFVKELAYKGHEIASHTVNHRLPPGTSSADYPEAVAEITGMREKVYQGTGDKAISDKMVGLRSPYLLVAHNVQFDALKNNGFLYDTSITNIETSSGRP
PLWPFTLDYIMPTCPNVPCPTKPYPGLWEVPINGFVGSNNYGCSMIDTCSVGSNVFTATKQEWYTFFQKNFDYFNPTKVPMHLFTHGSMFLRSPPSFEALVSWFQDLQL
NRKDVWIVTPTVIEWMKKPLTNSEMIAQKWGCN

**Contig 936 – Mucin 17 isoform X10 (1.88e-120)**

MANLGVSPPLVLGLILWLTTQAASVSHKCSSKNANSANVQTLVAPCRTTIDQQIDGYYMSADDYDRYNDNNVPDGNCTFVKSYESTVCCPGWTGSNCTTPNCNPPCQNG
GICKENDNNPYGSPVCICPTPFSGVRCEEKKTALLQEPGKKYCYRGESCDGGLTTSSTTSYGDCCNNKMTGSWGSQSSKCTLCRPTVTRDSTNGKVTCAALGSHDYRTF
DSVSYKFTSLCGVPLFIVPGLQIYVVSECDVKNKCTCFKKVFIFTKDFTYVIGGSSLTKVRSDGVPPIETHSIKDMKTTTFIDELKLKYTLERRAMYVAMTNNELEIRV
DDDGTVLLTIPSSSSWAGKITGACGNADKNPQDEAAYTTQTGANAMADRNKNPNIPCGVDLVACKPADVNANSACAPLYDFKSCHNSVPVEDFLSRCQSLYCTAVYKGS
ISSAQTAVCNLFSQYHTLCFLKKSETFPWQKNTLCPVRCPEPKVFDPFIRSRCPLTCGSSQQAYAHINCDTEPYAGCVCPDGYAKFESGCVPANDCPCVGSDGNKYKRG
EKYGVPGKCDVCVCDDNGLWKCKETSTVCIRSCRIFGVRNFITFDNRQFIIEDASGKLVLLQFGNNSVTIEKGNADGASADATIAFTVTVNWNGQQSSIQVSGVGATLQ
NAVTHDTVISQVSSNYYLVETANGKIRVLMSRTGLTQVDA

**Contig 2928 – Metalloproteinase-like ADAM family Mig-17 (3.13e-111)**

MKLSVYCAIAAMALSANMCAGETYVAEGYFVMDSKAVEIYINEIQGTDTYENKKAQAIARVREDVNYILSETNKLFGSLNAYGLDIQISVRKDILDTSLFPESALQNGN
IISDRISQQLFSDWLRLQESYTNYKFDFALLWTGYDIYGPSGVYTTAIANSGKVCDPILATSVVEFNATYSTVVTTAHTIALVLGASYDGVASDQIMAASNSPLHKRRW
SFSECSANDIKEFFGLVTPICLQTTNPASTTPASSWDTYNGRLFDSNAMCKRAINDVRSYACLSSNLYNKLTAKGDQLCQKTYCLLPESNLCLATYTSDGHICDYKKRC
DKGKCVTDTSAASANVDPECVLGDQEVVFFPKVPFKGTCQQFIAQQGKAMCYNSIIKDVCCATCKSYRNGPTDCEYGDKNDLCP

**Contig 3761 – Peptidoglycan domain containing protein (1.82e-72)**

MGSFDFIVFVVSLVAVVSADITTSTVNNGDCVCATTNVNARNRAGLSGTVRLTISSGQCVTASGASRRANGYKWYQVTHSGQTLWVAGSYLSQASASSCSPAPSGGSCP
SVQQLACNLLNNSKLTLATAHASGRSDNAFARNNLRDTCDGGQAALSRYTCRECRNPGAPGGSVCIAESVLRYLAVLVSRGRVKVNEIAGACHSCSSKHYQGLAIDLDD
GPRDQEFVTTCNNRGGRGINEGNHIHCQFNS

**Contig 129 – Epiphragmin *Cernuella virgata* (0.0)**

MASLSWILLLSAVPCITGIQFYITRSPGHENCALTECSYDATEHESDKFSIKQISIWDVTQTSNKYQLAYRKFGDKDTYFDPTKLNDIKGTSKVTKFHADIRLNFRNPA
DCMYESYECYVVYVDDLGKKQIVRKTIAPDSPDVDNDCSCSAITYHLDKLTNRANGASRQLYGLSLNLDNIQVDLNSVTKGTTELSSDIARLQESHQNILVSVNELRES
GREIDSLMNNLRSTDYTVESKINELLNLDEDKKQTIAALRQNDKNTDKQLSQLTRDDNSQMTSIELLQQADGVALTDINNLQQRDDNLEAKVNSLKKESSSLTNQIGQL
RENDNSARADISVLSSHDGEIQGATAGLRKTDNEIDRQISQLNRNDVELLEDAAQAKQATADKVREINELGQKFSDIGADIARLEQNDQDIDAKLDGLDKKNAEIDNDL
NDLLSEDVKIREAIDERKTVNAVHQESIGYLRRKNTAFQLQISDLLVTDRVIVTQITEVSSIGSSVQSSVQVAQVQQRDIREEMDRLENLMRVIESRIRTATITTTSET
TSSLVLTSCTRGMPHDNARAAILLYGSVPTVCDTETDGGGWILFQRRTNGEENFYRTWSDYKVGFGNPSGNFYIGNDNLAKLTDAGFTELRIDLTYKGKQYYAHYSQFR
VEGEESKYRLHVSGYEGSAPDSFSYHDGSQFSTYDQDNDGSYDSCAKSKKGGWWYKDCYESNLNGVWGKDDDTGMIWKAFSDDKSVSVVEMKPINYA

**Contig 199232 – Perlucin-like protein (1.74e-13)**

MMSRMMLFLTLAFTAVAVDAVNGYPSTDCWPGWIYFSGSCYAFADVPLWWDAASMCNANGGYLVEIETAAENNWLVARLRSSNYGSVWTGASESLHRGTFLWGKSGRAL
TYTDWSNI

**Contig 77247 – Cystatin-like protein *Biomphalaria glabrata* – (6.43e-71)**

MKVLLLIATALTFITGERLVGGKQPYDATLADPFVTFAINRINEFYSANGDQRARTGIRIVSATSQIVAGVKYNFKIEVTGGNVNEVCNVEVWSRQWLPEPENTQLIGQ
PTCAPKV

**Contig 235341 – Uncharacterised protein *Pomacea canaliculata* (6.0e-48)**

MLQDVPAWAGITALLVCIYVSAEAASQAIYMTNYCSMTVSVHGSLLLELTPPDSLHDERSGDFLHCDVDIVAPRSMKLLTHVQQLGISSAADHSDRLHLYEGNPGNTRL
TPIKGLYGRLDTSLSISTPSGRKVKDVRTLGNKLKVDYLGKPTIRTPGFRLIITTFQDPTSAGKCPPKYYHCTRASVCVMRHVVCDGNPNCGKDDDDDERECDSKDTVT
NLLAKYSLTRDIIVVILLPVVTFLAAVTAIYLAARQYIRHRIDVPEICAVHFTAKKDGCVDIAAKHGQYNPPTYQDIMDMCSGYEPPPTYNQLTTSVTRCDRSATKCDS
SATTGDRSVGISNGRCVLKSEVRRLSRECAQCGRVTRGFRISVIPEEDECCRHDDDVLHSVSDWDDNEEFQTRRGKREKGKVKKKIKYSENASCIDETKDDFREDNQNS
FMASCVVCCAELCHSQEGHTSTHMSHVTPSCRRGLACSKCTQPHDKRYKRSSNLAKPHVNIYSSYVDFTHPQEES

**Contig 62832 – Novel**

MLFASYATALLFVCAFESLAAFQIRESVKDTSDDVGKGPIELANDVISTLGKLSEGRSGAEVKADLEEIVQLASDLLEDTHEDVQKVPAPTNDGDVSKKTAIEAAVKLN
GELFLLSRHARKLLTAVENRLAEETVLNIIAIKQRASSVISIIDESGIDFVSYQDPFETAQGSRKRRRNKCDGISFWDVCVGHRH

**Contig 7487 – Novel**

AMRVVVVAAVLLILLAFVVGLESAPSARDEAEARAAVERQVKRELDAFERALFADDDHEVRYRRGWFKKLKKKLKNWWNKNRGQVVVSTAVQAGAKYLGGKRDAADFQW

**Fig 2. Annotation of full-length secreted proteins identified in trail mucus of *C. aspersum*.** Yellow shading, cysteine residues; Blue shading, N-glycosylation sites; Green shading, signal sequences; Red shading, putative cleavage sites.

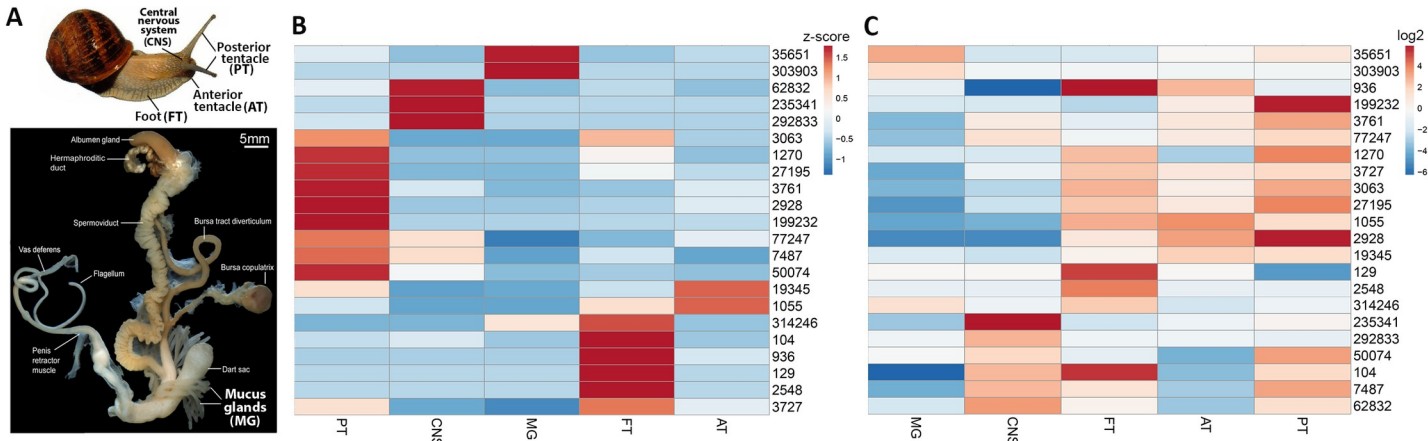

**Fig 3. Relative gene expression of secreted proteins in Foot (FT), Mucous Gland (MG), Central Nervous System (CNS) and Posterior (PT) and Anterior Tentacles (AT) of *C. aspersum*.** (A) Location of relevant snail tissues and abbreviations. (B) Relative gene expression based on z-score. Red indicates highest expression levels of gene as compared to other tissues. (C) Relative gene expression based on log2. Red indicates higher expression of gene as compared to other genes.

expect that *C. aspersum* also contains more perlucin-like genes, however, only one may be required for its trail mucus. A comparative sequence analysis of the clade containing *C. aspersum* perlucin-like proteins, demonstrates that besides the cysteines, several residues are highly conserved, including tryptophan, serine, glutamic acid and aspartic acid (Fig 4B).

A large number of trail mucus proteins annotate to well-known structural proteins, including collagen and actin. Collagen is a protein found widely across the animal kingdom [54] and is the main contributor to the structure of connective tissue [55]. Interestingly, variants of collagen have been detected in external secretions, such as the cocoon silk of the willow sawfly, *Nematus oligospilus* [54]. Given that the snail's muscular foot helps to secrete the trail mucus, it is not surprising to find an abundance of this protein in the trail mucus. In addition, investigations into other organisms, such as the mussel *Mytilus edulis* have detected the presence of collagen outside the living tissue [56], suggesting that it plays a role in mucus structure. Collagen could be a contributing factor for the healing properties ascribed to snail mucus and popularity in the cosmetic industry [57]. Similarly, the oxygen-binding protein haemocyanin was found in the trail mucus. Haemocyanin is a major component of the haemolymph [58] that can be found in the haemocoel of molluscs. Molluscan hemocyanin has also been shown to have antibiotic activity [59–61], which could account for this observation of snail mucus in previous studies [62–65]. Another study found that haemocyanin subunit-1 had an affinity for binding with a water-borne pheromone in the freshwater prawn *Macrobrachium rosenbergii* [66], suggesting that this protein may have a role in chemical communication in the snail.

It is possible that one or more of the novel trail mucus proteins identified in this study could function as a pheromone, or as a component of a pheromone blend, as in the case of the *Aplysia* aquatic attraction pheromone [30, 32, 33]. Protein pheromones have also been identified in a number of terrestrial species that encompass a range of phyla, including Annelida [67, 68], Chordata [69] and Arthropoda [70]. However, further work needs to be undertaken in order to establish snail response to these putative pheromones, and whether they function alone or as a component of a mixture. A response may also be dose-dependent, necessitating experimental trials testing a range of concentrations. As mucus for the SDS-PAGE analysis was collected toward the end of the breeding season, further work should compare seasonal differences in mucus peptides, along with differences between juveniles and adults. Such investigations would help to narrow down a putative reproductive pheromone, which could then be

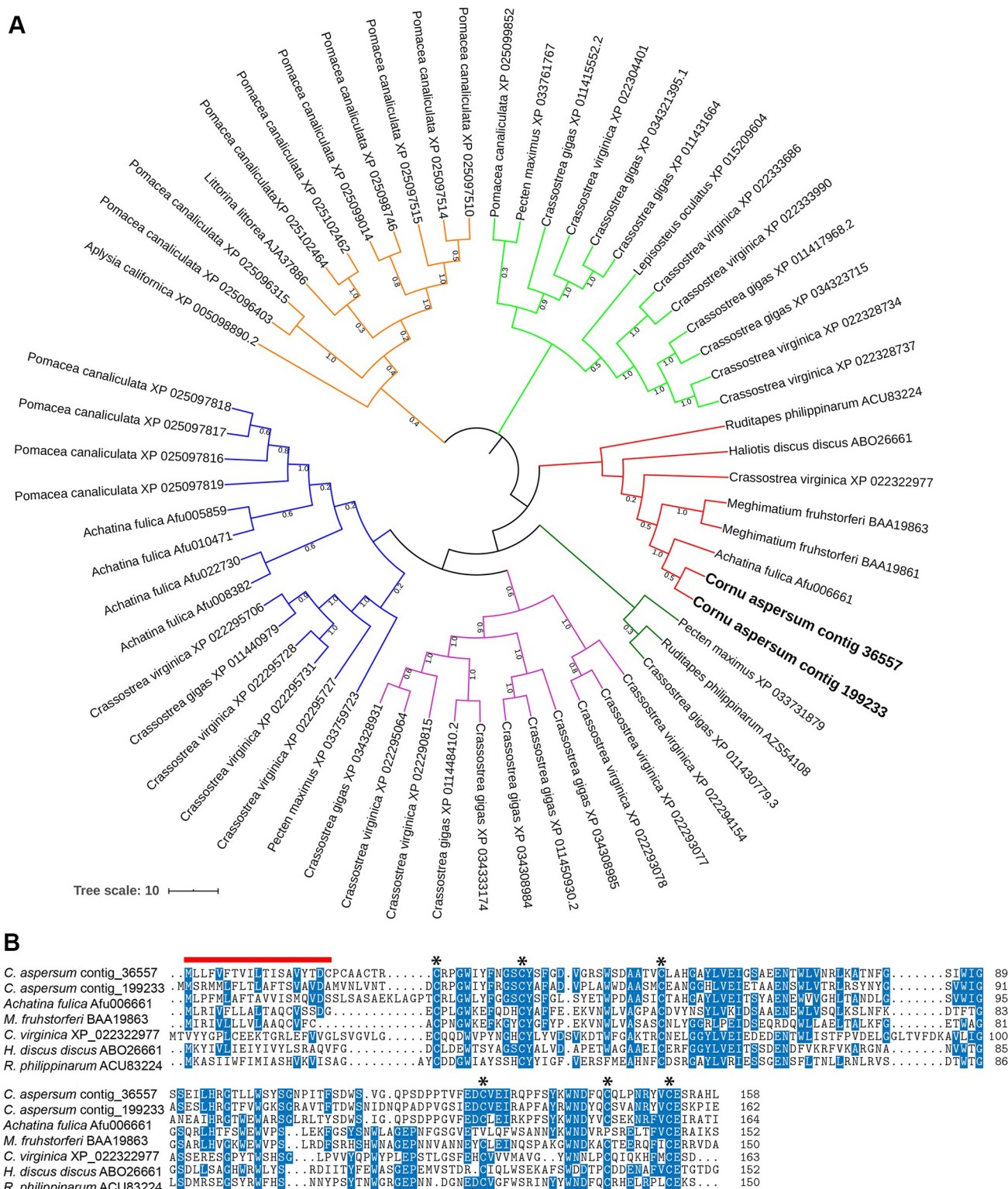

**Fig 4. Comparative analysis of perlucin-like proteins.** (A) Phylogenetic tree showing 6 clusters of perlucin-like proteins (colour-coded), including two found in *Cornu aspersum* (red cluster). (B) Multiple protein sequence alignment of mature perlucin-like proteins represented in phylogenetic cluster with *Cornu aspersum*. Shading and sequence logo (above alignment) provides level of conservation between species.

tested in a behavior bioassay to determine snail response and potentially utilised in development of a control strategy. It should also be noted that the SDS-PAGE method led to a much higher number of proteins identified than the Sep-Pak method (145 versus 30, respectively); however, this may be due to the higher number of snails that were sampled in the former, with the larger sample size leading to a higher protein diversity.

Of the novel proteins, contig 62832 was expressed relatively highly in the CNS, suggesting a neuropeptide. Neuropeptides can function in a wide range of physiological process, including reproduction. Contig 7487 was expressed relatively highly in the posterior tentacle (PT). The dibasic cleavage sites in this protein are suggestive of bioactivity. Further work should include synthesis of proteins and behavioural testing to determine snail response and elucidate protein function.

## Volatile analysis of trail mucus

To determine what volatile substances were emitted from the mucus trail, headspace volatile extraction using thermal desorption was performed. An example chromatogram showing the most significant peaks produced by GC-MS of a sample of trail mucus is shown in Fig 5. Fig 6 provides a list of the compounds consistently appearing in highest abundance over all mucus thermal desorption experiments, after removing compounds common to the negative control (e.g. siloxane).

The VOCs that were collected onto the thermal desorption tubes are also worthy of further investigation in regard to their role as putative pheromones, or components of a pheromone blend. Propanoic acid is known to be produced by bacteria of the *Staphylococcus* genus [71], of which at least two species are present in trail mucus based on a microbial diversity profile (K. Ballard, unpublished research). In the house cricket, *Acheta domesticus*, propionic acid was isolated from the excreta of adults and promoted aggregation of their larvae [72]. This example

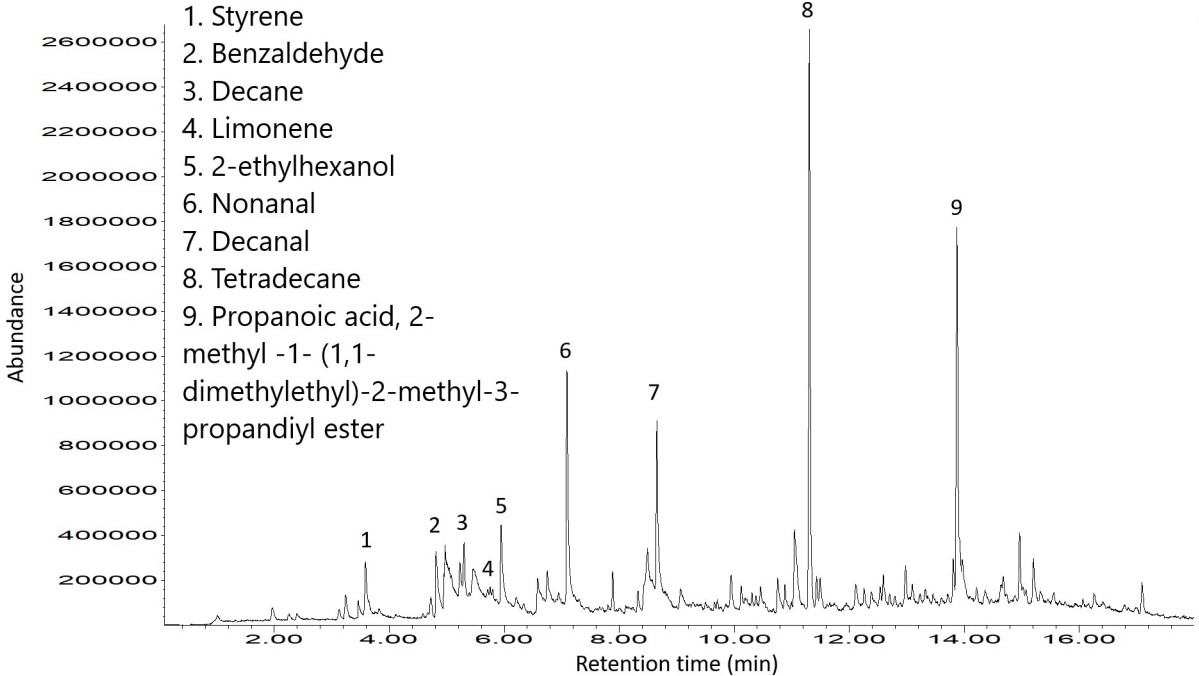

**Fig 5. Total ion chromatogram of mucus sample with the most frequently occurring compounds labelled.** Compounds are: 1 –styrene, 2 –benzaldehyde, 3 –decane, 4 –limonene, 5–2-ethylhexanol, 6 –nonanal, 7 –decanal, 8 –tetradecane, 9—propanoic acid, 2-methyl-1-(1,1-dimethylethyl)-2-methyl-3-propandiyl ester.

| Name | Retention time (min) | Formula | Structure |
|---|---|---|---|
| Styrene | 3.6 | $C_8H_8$ | |
| Benzaldehyde | 4.8 | $C_7H_6O$ | |
| Decane | 5.2 | $C_{10}H_{22}$ | |
| Limonene | 5.8 | $C_{10}H_{16}$ | |
| 2-ethylhexanol | 5.9 | $C_8H_{18}O$ | |
| Nonanal | 7.2 | $C_9H_{18}O$ | |
| Decanal | 8.8 | $C_{10}H_{20}O$ | |
| Tetradecane | 11.3 | $C_{14}H_{30}$ | |
| Propanoic acid, 2-methyl-1-(1,1-dimethylethyl)-2-methyl-3-propandiyl ester | 14.0 | $C_{16}H_{30}O_4$ | |

**Fig 6. Most abundant VOCs detected in *C. aspersum* trail mucus.**

demonstrates that not only can propanoic acid function as a pheromone, but its presence and abundance could also potentially be influenced by the microbiota of the organism.

Limonene is another interesting VOC discovered in the trail mucus. Limonene is a plant-based monoterpene which is found at high levels in citrus fruit peels [73, 74]. Recent research has explored the effects of limonene on insects. One such study found that the addition of the monoterpenes limonene and α-pinene to the synthetic pheromone of the Northern Spruce Bark Beetle, *Ips duplicatus*, enhanced the response of beetles to the pheromone [75]. Another study found that reduction of limonene production in citrus trees led to fruit being less attractive to the fly *Ceritilus capitata*, and less susceptible to infection by the fungus *Penicillium digitatum*, implying that limonene might be a plant adaptation to attract organisms that facilitate seed dispersal [75] or confer some sort of protection or benefit to the plant. As snails were fed on carrot and lettuce it is interesting that this terpene would be found across different snail samples. However, the concentration was higher in snails that had been recently taken from a natural environment. Given that it has been shown to affect the response of other invertebrates, it is worthy of further investigation to determine its consistency in snail mucus and whether it may be a component of a pheromone blend. This also suggests that the environment and food sources have an effect on the components of mucus. Limonene has also been shown to have anti-tumour properties in a variety of cancers, including lung and breast cancer [76–78]. The presence of this substance in snail mucus may be a possible explanation for the anti-cancer properties exhibited by snail mucus in other studies [79, 80].

The presence of benzaldehyde in the trail mucus also warrants further investigation. Chemotaxis in response to benzaldehyde has been exhibited in the model organism, *Caenorhabditis elegans* [81]. Interestingly, *C. elegans* shows an initial attraction to benzaldehyde, followed by an aversion response after an hour of exposure. This observation highlights the complexity of olfactory responses, which may be dose and time-dependent and can be affected by environmental conditions [82]. In addition, previous work demonstrated that following exposure to a high concentration of benzaldehyde, attraction to low concentrations was reduced [83]. This premise is applied to the control of some agricultural insect pests [12, 13], in order to reduce male response to female pheromones. Olfactory receptors in another model organism, *Drosophila melanogaster*, also show a significant antennal response to benzyl alcohol, a precursor of benzaldehyde [84]. This particular study found increased responses to all compounds containing a cyclic ring, providing a promising direction for olfaction research in other invertebrates. Somewhat surprisingly, both limonene and benzaldehyde were found to be components of the defensive secretion of the stick insect *Sipyloidea sipylus*, which were effective in deterring rats [85]. As this species is considered to be primarily ground-dwelling, with a similar range of predators to terrestrial snails, this highlights the possibility that these volatiles may function in defense, to deter potential predators from the trail.

Ethyl hexanol is widely used as a fragrance in the cosmetic industry, and forms esters with emollient properties [86]. However, this compound can also be produced by the hydrolysis of the common plasticizer, diethylhexyl phthalate, particularly in a damp or humid environment [87]. Therefore, this volatile may be a product originating from the plastic tubing used to collect direct airflow in this experiment. In addition, a large number of peaks in the chromatogram were not identified, and it is possible that one or more of these could be important in communication. Along with replication, a deeper analysis of the detected volatiles could yield a more comprehensive result and allow for more targeted testing.

Interestingly, there were no volatiles found in common with the study of Sallam et al. [24]. This may be due to different snail species, different diet and environment, or different method of collection, which was not clearly elucidated in that study. Further study would investigate the volatile components of other invasive land snail species for comparative purposes.

Additionally, utilizing an alternative method of collection, such as solid-phase microextraction (SPME), would provide stronger evidence for the presence of the volatiles detected in this study, as well as reducing the risk of inclusion of contaminants. Behavioural assays using an olfactometer to assess snail response would help to determine if any of these volatile substances would be useful as a deterrent or as bait in a pheromone trap. Alternatively, one or more of these chemicals might be utilized in the interference of conspecific attraction. It should also be noted that the trail mucus collected in this study for VOC extraction was collected between late May and early June, which is outside the breeding season for this species. As trail mucus composition may differ between the snail's hibernation period and the active reproductive period, further investigation of VOCs during the reproductive season would be likely to yield additional molecules or adjusted concentrations of known volatiles.

*Cornu aspersum* is generally a solitary snail; therefore, there may be no need to secrete pheromones when in close proximity to other conspecifics. However, as several proteins did not significantly match to any other sequences in the non-redundant protein sequence database, it is quite possible that one or more of these proteins might contribute to a pheromone, or pheromone blend. These results highlight the complicated nature of snail communication, which is likely highly contingent upon environmental influences and physiological status. Results of this study pave the way for future studies of this kind, which may help to determine which proteins are consistently present, and which are changeable as a result of environmental or internal influences. In many areas of the world, more serious problems are posed by related species, including *Theba pisana*, the White Mediterranean Snail. This species is a major pest of cereal crops and has caused extensive damage and economic losses in the wheatbelt of the Yorke Peninsula in South Australia [6, 7] and is also present in Western Australia, Victoria, and some areas of New South Wales and Tasmania [8]. The grape industry is also a victim of invasive snails, particularly *C. aspersum* and *T. pisana*. Not only do the juveniles of both species consume the young leaves and buds of the vines, but mucus trails on the fruit reduces aesthetic appeal and therefore the value of the fruit [4].

Trail mucus is an important secretion for land snails that has not been rigorously investigated to date, apart from several investigations into its antimicrobial properties [62–65]. The results of the present study demonstrate that the small percentage of mucus that is not water is host to a variety of biomolecular components, including proteins and volatile metabolites. This study helps support a role for trail mucus in conspecific chemical communication. This study has provided a starting point for further investigation into the components of snail mucus, and which of these might function in conspecific communication. Further work should explore the differences in mucus composition over different seasons, different environments and different life stages. This should focus primarily on the snail's breeding season, when the snail is more active, and components are likely to be more varied and abundant. Furthermore, differences in mucus composition between seasons could assist in identifying a putative pheromone based on molecules that are present in the active breeding season, and absent in the hibernation season. Such investigations could also explore the possibility of non-volatile small molecules that may function in communication. Ongoing research should include exposure to mucus from other snail species to determine if responses are species-specific. A more comprehensive knowledge base of snail chemical communication via the mucus trail could lead to potential applications in the fields of agriculture, ecology and medicine.

## Supporting information

**S1 File. All individual sequences identified in trail mucus of *C. aspersum*.**
(TXT)

**S2 File. Remaining signal peptide sequences identified in trail mucus of *C. aspersum*.**
(DOCX)

## Acknowledgments

Thanks to Shahida Mitu for lab assistance and computer support.

## Author Contributions

**Conceptualization:** Kaylene R. Ballard, Scott F. Cummins.

**Data curation:** Kaylene R. Ballard, Anne H. Klein.

**Formal analysis:** Kaylene R. Ballard, Richard A. Hayes, Tianfang Wang, Scott F. Cummins.

**Funding acquisition:** Kaylene R. Ballard, Scott F. Cummins.

**Investigation:** Kaylene R. Ballard, Richard A. Hayes, Tianfang Wang, Scott F. Cummins.

**Methodology:** Kaylene R. Ballard, Richard A. Hayes, Tianfang Wang, Scott F. Cummins.

**Project administration:** Kaylene R. Ballard.

**Resources:** Richard A. Hayes.

**Software:** Anne H. Klein.

**Supervision:** Richard A. Hayes, Scott F. Cummins.

**Validation:** Kaylene R. Ballard, Scott F. Cummins.

**Writing – original draft:** Kaylene R. Ballard.

**Writing – review & editing:** Kaylene R. Ballard, Anne H. Klein, Richard A. Hayes, Tianfang Wang, Scott F. Cummins.

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
