## [Decision Letter · Decision Letter 0]

26 Mar 2021

PONE-D-20-38531

The protein and volatile components of trail mucus in the Common Garden Snail, Cornu aspersum

PLOS ONE

Dear Dr. Cummins,

Thank you for submitting your manuscript to PLOS ONE. After careful consideration, we feel that it has merit but does not fully meet PLOS ONE’s publication criteria as it currently stands. Therefore, we invite you to submit a revised version of the manuscript that addresses the points raised during the review process.

This is an interesting and important study of potential chemical signals used in communication of an economically important snail species.  While it is well written and the experimental approach sound, the reviewers identify several possibilities for improvement.  The authors should respond to the comments provided especially the suggestion to test the effect on heart rate of mucous from other snail species in order to provide insight on the specificity of the response.

We look forward to receiving your revised manuscript.

Kind regards,

Joseph Clifton Dickens

Academic Editor

PLOS ONE

Additional Editor Comments:

This is an interesting and important study of potential chemical signals used in communication of an economically important snail species. While it is well written and the experimental approach sound, the reviewers identify several possibilities for improvement. The authors should respond to the comments provided especially the suggestion to test the effect on heart rate of mucous from other snail species in order to provide insight on the specificity of the response.

Journal Requirements:

3. We note that you are reporting an analysis of a microarray, next-generation sequencing, or deep sequencing data set. PLOS requires that authors comply with field-specific standards for preparation, recording, and deposition of data in repositories appropriate to their field. Please upload these data to a stable, public repository (such as ArrayExpress, Gene Expression Omnibus (GEO), DNA Data Bank of Japan (DDBJ), NCBI GenBank, NCBI Sequence Read Archive, or EMBL Nucleotide Sequence Database (ENA)). In your revised cover letter, please provide the relevant accession numbers that may be used to access these data.

For a full list of recommended repositories, see http://journals.plos.org/plosone/s/data-availability#loc-omics or http://journals.plos.org/plosone/s/data-availability#loc-sequencing

 'Grains Research Development Corporation, Contract code is USU1903-001RSX]'             

5. Thank you for stating the following in the Financial Disclosure section:

 'Grains Research Development Corporation, Contract code is USU1903-001RSX'  

We note that you received funding from a commercial source: Grains Research Development Corporation

Reviewers' comments:

Reviewer's Responses to Questions

**Comments to the Author**

1. Is the manuscript technically sound, and do the data support the conclusions?

Reviewer #1: Yes

Reviewer #2: Yes

2. Has the statistical analysis been performed appropriately and rigorously? 

Reviewer #1: Yes

Reviewer #2: Yes

3. Have the authors made all data underlying the findings in their manuscript fully available?

Reviewer #1: Yes

Reviewer #2: Yes

4. Is the manuscript presented in an intelligible fashion and written in standard English?

Reviewer #1: Yes

Reviewer #2: Yes

5. Review Comments to the Author

Reviewer #1: The authors provide foundational knowledge towards characterizing the physiological relevance and chemical identity of an economically important snail’s trail mucus. This research is a logical first step towards elucidating the potential role of snail mucus in communication between conspecifics. These results will inform future studies aimed at developing novel pest control strategies for snails.

The manuscript is very well written. The rationale presented in the introduction is convincing, noting that I am not well versed in the chemical ecology of gastropods. I was left wondering if non-protein components, like those synthesized in mucous-producing cells, may be important for contact sensation via the mucous. I realize that this may be beyond the scope of this study, but it may be worth mentioning in the introduction or discussion.

Methods: Do we know the active ingredient in On-guard Snail Gel? The methods section looks very complete. I have no concerns here.

Results and Discussion: It would be great to measure heart rate responses of these snails to mucus from a closely related species and that of a more distant relative. The positive and negative controls used here work well to show there is a response but do little to address the nature of this response, which the authors acknowledge.

The tangential discussions of identified mucous proteins is fantastic. Thank you for including this information.

I think it would be helpful to readers to use the full name of each tissue on the bottom axis of the heatmap in Figure 3 for quicker reference. It looks like you should be able to fit them into the space available, perhaps at an angle. Likewise, you could insert the corresponding chemicals in Figure 5 at each peak.

I only noticed one typographical error: Line 108 –punctuation error.

It could increase visual interest to include one diagrammatic image or labeled photograph of the relevant snail anatomy. Though very basic, this could help gastropod novices become more connected to the work. This could even be a figure inset.

This is a great submission. Congratulations.

Reviewer #2: This paper provides useful information on the common garden snail’s mucus and has the potential to help identify signaling compounds. Given the economic importance of these animals as agricultural pests, this is a worthwhile contribution. I have some suggestions that I hope will improve this manuscript. If the authors can address these points, I feel that the manuscript is worthy of publication.

When the authors measured the effect on the heart rate of contact with mucus, it would be interesting to look at one or more mucus samples from distantly related organisms, rather than just water as a negative control. Are the snails detecting common mucus molecules, or something unique to that species?

What were criteria for selecting the proteins in Fig. 2? The authors choose to present 10 of the 22 proteins that were predicted to be secreted. Why were these 10 selected and not the other 12? Were they more abundant, or was the selection arbitrary? Notably, two of the contigs that had a markedly higher relative expression in foot (104, 2548) are not included in this ten; why not?

In figure 3, the authors present relative expression compared to other tissues. It would also be interesting to provide the TPM values to give a sense of which were the most highly expressed proteins in the mucus.

Lines 160-162 state, “A transcriptome-derived protein database was prepared by combining the foot and anterior tentacle with tissue transcriptomes of the mucous gland, central nervous system (CNS) and posterior tentacle using the CLC genomics Workbench.” It is not clear at this point where the non-foot transcriptome data comes from. Later in the paper it is identified, but it should be made clear here. Additionally, the methods only state that foot tissue was used, but later in the methods it is stated that anterior tentacle was also used. This should be cleared up.

Line 172 states, “Trail mucus biomolecules were isolated using Sep-Pak plus C18 cartridges (Waters) according to the manufacturer’s instructions.” Give more information on this separation. Was this FPLC? What was the buffer system? Liquid chromatography of snail mucus is challenging due to the large glycosaminoglycans, which have a tendency to clog columns.

For the second extraction method, were all visible bands analyzed by LC-MS/MS? If not, what were the criteria for inclusion?

Were there notable differences in the proteomes derived from the two extraction methods?

Note that C-lectins with similarity to perlucin have also been identified as centrally important in the adhesive properties of a terrestrial slug’s defensive mucus (Smith et al., 2017, “RNA-seq reveals a central role for lectin, C1q and von Willebrand Factor A domains in the defensive glue of a terrestrial slug” Biofouling).

Fig 3. Provide some guidance for interpreting Z-score in the figure legend. I assume that red indicates higher relative expression. Not everyone will be familiar with Z-score.

The last page and a half of the discussion could be more focused. There is some repetition that could be removed.

6. PLOS authors have the option to publish the peer review history of their article (what does this mean?). If published, this will include your full peer review and any attached files.

Reviewer #1: No

Reviewer #2: No

---

## [Author Response · Author response to Decision Letter 0]

26 Apr 2021

Response to Reviewers for Manuscript PONE-D-20-38531

Editors Comments

1. Style requirements met

2. Reference list correct

3. Genbank accession numbers have been uploaded to NCBI and noted in manuscript

4. Added to cover letter

5. Added to cover letter

Reviewer 1

The manuscript is very well written. The rationale presented in the introduction is convincing, noting that I am not well versed in the chemical ecology of gastropods. I was left wondering if non-protein components, like those synthesized in mucous-producing cells, may be important for contact sensation via the mucous. I realize that this may be beyond the scope of this study, but it may be worth mentioning in the introduction or discussion.

Response: Added to discussion (Line 514).

Do we know the active ingredient in On-guard Snail Gel? The methods section looks very complete. I have no concerns here.

Response: In spite of examination of the label and an email to manufacturer (in UK), the active component has not been confirmed. As the product is stated as being non-toxic and natural, it is likely that it contains a combination of natural ingredients that the snails find aversive.

I think it would be helpful to readers to use the full name of each tissue on the bottom axis of the heatmap in Figure 3 for quicker reference. It looks like you should be able to fit them into the space available, perhaps at an angle. Likewise, you could insert the corresponding chemicals in Figure 5 at each peak.

Response: Chemical names have been included within the chromatogram to ensure easy reference while maintaining tidiness and clarity.

It could increase visual interest to include one diagrammatic image or labelled photograph of the relevant snail anatomy. Though very basic, this could help gastropod novices become more connected to the work. This could even be a figure inset.

Response: To help readers comprehend the anatomy, we have added an anatomical figure (as 3A), inclusive of abbreviations used in heatmaps 3B and 3C.

I only noticed one typographical error: Line 108 –punctuation error.

Response: Corrected.

Reviewer 2

When the authors measured the effect on the heart rate of contact with mucus, it would be interesting to look at one or more mucus samples from distantly related organisms, rather than just water as a negative control. Are the snails detecting common mucus molecules, or something unique to that species?

Response: Agreed, and are indeed exploring this type of analysis between related species within the scope of another research project performed by the first author, i.e profiling mucus molecules between sympatric pest land snail species.

What were criteria for selecting the proteins in Fig. 2? The authors choose to present 10 of the 22 proteins that were predicted to be secreted. Why were these 10 selected and not the other 12? Were they more abundant, or was the selection arbitrary? Notably, two of the contigs that had a markedly higher relative expression in foot (104, 2548) are not included in this ten; why not?

Response: The 10 that are presented were thought be the most likely to play a role in communication and are generally of more interest. Eg. Novel or uncharacterised. The remaining 12 contain sequences that match to several variants of collagen and another variant of epiphragmin. These sequences are available in the supplementary material (S2).

In figure 3, the authors present relative expression compared to other tissues. It would also be interesting to provide the TPM values to give a sense of which were the most highly expressed proteins in the mucus.

Response: This is the purpose of heatmap Fig 3C, which has now been more clearly explained in the figure legend.

Lines 160-162 state, “A transcriptome-derived protein database was prepared by combining the foot and anterior tentacle with tissue transcriptomes of the mucous gland, central nervous system (CNS) and posterior tentacle using the CLC genomics Workbench.” It is not clear at this point where the non-foot transcriptome data comes from. Later in the paper it is identified, but it should be made clear here. Additionally, the methods only state that foot tissue was used, but later in the methods it is stated that anterior tentacle was also used. This should be cleared up.

Response: Apologies for the oversight – the anterior tentacle was removed and RNA isolated at the same time and using the same methodology as the foot tissue. This has been added to the methods. (Line 152).

Line 172 states, “Trail mucus biomolecules were isolated using Sep-Pak plus C18 cartridges (Waters) according to the manufacturer’s instructions.” Give more information on this separation. Was this FPLC? What was the buffer system? Liquid chromatography of snail mucus is challenging due to the large glycosaminoglycans, which have a tendency to clog columns.

Response: Apologies for the oversight – The use of 100% acetonitrile for cartridge preparation and 60% acetonitrile for elution has been added to methods (Line 176). This pre-preparation, prior to LC-MS, is an approach that would remove large glycosaminoglycans.

For the second extraction method, were all visible bands analyzed by LC-MS/MS? If not, what were the criteria for inclusion?

Response: Yes, all visible bands were subjected to in-gel trypsin digestion and analysis and this information has now been added to methods (Line 184 – ‘All visible bands were excised….).

Were there notable differences in the proteomes derived from the two extraction methods?

Response: Yes, and this information has now been added to the manuscript (Line 396).

Note that C-lectins with similarity to perlucin have also been identified as centrally important in the adhesive properties of a terrestrial slug’s defensive mucus (Smith et al., 2017, “RNA-seq reveals a central role for lectin, C1q and von Willebrand Factor A domains in the defensive glue of a terrestrial slug” Biofouling).

Response: Thank you for drawing our attention to this very interesting paper. A reference to this paper has been added to our manuscript (Line 348).

Fig 3. Provide some guidance for interpreting Z-score in the figure legend. I assume that red indicates higher relative expression. Not everyone will be familiar with Z-score.

Response: Apologies for the confusion. Yes, red indicates higher expression and this information has now been added to the Figure 3 legend.

The last page and a half of the discussion could be more focused. There is some repetition that could be removed.

Response: Agreed. This part of the discussion has been revised and we now feel it is more focused.

---

## [Editor Report · Decision Letter 1]

29 Apr 2021

The protein and volatile components of trail mucus in the Common Garden Snail, Cornu aspersum

PONE-D-20-38531R1

Dear Dr. Cummins,

We’re pleased to inform you that your manuscript has been judged scientifically suitable for publication and will be formally accepted for publication once it meets all outstanding technical requirements.

Kind regards,

Joseph Clifton Dickens

Academic Editor

PLOS ONE
---

## [Editor Report · Acceptance letter]

3 May 2021

PONE-D-20-38531R1 

­­The protein and volatile components of trail mucus in the Common Garden Snail, *Cornu aspersum*

Dear Dr. Cummins:

I'm pleased to inform you that your manuscript has been deemed suitable for publication in PLOS ONE. Congratulations! Your manuscript is now with our production department. 

Kind regards, 

on behalf of

Dr. Joseph Clifton Dickens 

Academic Editor

PLOS ONE